# The Role of Mechanoperception in Plant Cell Wall Integrity Maintenance

**DOI:** 10.3390/plants9050574

**Published:** 2020-05-01

**Authors:** Laura Bacete, Thorsten Hamann

**Affiliations:** Institute for Biology, Faculty of Natural Sciences, Norwegian University of Science and Technology, 5 Høgskoleringen, 7491 Trondheim, Norway; laura.bacete@ntnu.no

**Keywords:** cell wall, cell wall integrity, mechanosensing, mechanoperception, plant defense, plant environment interaction

## Abstract

The plant cell walls surrounding all plant cells are highly dynamic structures, which change their composition and organization in response to chemical and physical stimuli originating both in the environment and in plants themselves. They are intricately involved in all interactions between plants and their environment while also providing adaptive structural support during plant growth and development. A key mechanism contributing to these adaptive changes is the cell wall integrity (CWI) maintenance mechanism. It monitors and maintains the functional integrity of cell walls by initiating adaptive changes in cellular and cell wall metabolism. Despite its importance, both our understanding of its mode of action and knowledge regarding the molecular components that form it are limited. Intriguingly, the available evidence implicates mechanosensing in the mechanism. Here, we provide an overview of the knowledge available regarding the molecular mechanisms involved in and discuss how mechanoperception and signal transduction may contribute to plant CWI maintenance.

## 1. Introduction

The perception of plant cell walls by the scientific community has changed dramatically in recent years. Originally thought to be only inert structures surrounding all plant cells and providing both support and protection, they are now considered to be highly dynamic and plastic [1]. To date, targeted approaches trying to modify cell wall composition and structure to improve performance of food or bioenergy crops have had only limited success. Cell wall plasticity has been identified as one of the reasons for this, because it can apparently neutralize the effects of introduced genetic modifications to a significant extent [2,3]. These results imply the existence of a mechanism capable of detecting the effects of the introduced modifications and initiating compensatory responses to neutralize them [4]. The available evidence suggests that a key element underlying plasticity is the cell wall integrity (CWI) maintenance mechanism, since it seems to be capable of detecting effects of modifications and regulating responses. This mechanism is conserved throughout the plant kingdom, as molecular components implicated in it and their homologs have been described in both mono- and dicotyledonous plants as well as in liverworts like *Marchantia polymorpha* [4,5,6]. The CWI maintenance mechanism constantly monitors the functional integrity of cell walls and initiates compensatory changes in cell wall and cellular metabolism to maintain wall integrity during developmental processes like cell morphogenesis, as well as during exposure to biotic and abiotic stress. The available evidence implicates mechano- and turgor-sensitive processes in the early stages of CWI maintenance, thus suggesting that mechano-/turgor sensing may contribute to CWI maintenance.

Here, we briefly introduce relevant components of plant cell walls, followed by a short summary of the knowledge available regarding the CWI maintenance mechanism in *Saccharomyces cerevisiae* (*S. cerevisiae*), for the reason that the understanding of this simpler system can serve as food for thought regarding the processes in plants. More importantly, in recent years, additional detailed mechanistic information regarding the plant CWI maintenance mechanism has become available, indicating the usefulness of this approach, thus making an update of a previous review on the topic timely and relevant [7]. In the next sections, we present knowledge about the contributions of mechano- and turgor perception to CWI maintenance. This is followed by a critical assessment of the contributions of receptor-mediated CWI monitoring and wall-derived damage-associated molecular patterns (DAMPs). We close by discussing how the different mechanisms can contribute jointly to CWI maintenance, providing a short perspective on open questions and future challenges in this context. 

## 2. A Simplified Overview of Plant Cell Wall Composition and Structure

Plant cell walls are composed mainly of different polysaccharide types (cellulose, pectins, and hemicelluloses), lignin, and structural proteins, and can be divided into three major types with different composition and functions [8]. All plant cells undergoing developmental expansion have a dynamically modified primary cell wall consisting mainly of polysaccharides and proteins [8]. Cells that have completed their cellular expansion and are differentiating (e.g., to form vessels or fiber cells), deposit a secondary cell wall composed mainly of polysaccharides and lignin [9]. The third type of cell wall enables organ contraction during drought response, being particularly abundant in fiber-rich plants like flax and hemp, and consists of polysaccharides only [8]. Cell wall composition and structure vary widely between plant taxa. This is exemplified by the differences observed between type I and type II primary cell walls. Type I primary cell walls are found only in eudicots and non-grass monocots, while type II primary walls dominate in grasses [10]. Our current knowledge regarding composition and structure, as well as the processes giving rise to type I cell walls was recently summarized comprehensively [8], while type II walls were described in Reference [4]. Here, we focus on a subset of cell wall polysaccharides that have been previously implicated in CWI maintenance, and discuss processes taking place mainly in primary cell walls in *Arabidopsis thaliana*, since most of the available information derives from this model plant.

The main load-bearing elements in plant cell walls are the cellulose microfibrils, which consist of 18–24 β-1,4-linked glucan chains, each made of up to 10,000 d-glucose monomers [8]. The microfibrils are synthesized from uridine diphosphate–glucose by cellulose synthase (CESA) enzymes, which are localized in the plasma membrane and organized in cellulose synthase complexes (alternatively called rosette complexes) assembled in the Golgi apparatus. The residence time of CESAs at the plasma membrane is in the range of minutes while their half-life is longer, highlighting the involvement of tightly controlled exo- and endocytosis processes in cellulose biosynthesis [8,11,12]. Recently, several proteins have been implicated in these trafficking processes, providing insights into the underlying mechanisms [13,14]. CESAs are encoded by multi-gene families, and three different CESAs can normally be found in each cellulose synthase complex [8]. In *A. thaliana*, 10 CESAs have been identified, with CESA1, 2, 3, 5, 6, and 9 being active during primary cell wall formation, while CESA4, 7, and 8 mediate cellulose production during secondary cell wall formation [8]. Activity of the CESA proteins is regulated in different ways, including via phosphorylation, S-acylation, and interaction with associated proteins, exemplified by KORRIGAN and COBRA [15,16,17,18]. 

Pectins are a family of covalently linked galacturonic-acid-rich polysaccharides which constitute approximately 50% of the primary cell wall in *A. thaliana* [19]. In addition to their role as structural components, pectic polysaccharides also contribute to cell–cell adhesion and cell expansion [20,21], as well as having signaling functions during development and in plant immunity [22,23]. Their regulatory role in cell expansion seems to be brought about by the interaction of turgor pressure and their organization in a nanofilament structure in anticlinal cell walls [24]. They are produced in the Golgi bodies and modified during their incorporation into the cell wall at the plasma membrane [4]. While their chemical complexity implies a large number of enzymes being involved in their production, only a small number of enzymes have been characterized in detail [4]. The most abundant pectic polysaccharide is homogalacturonan, a linear homopolymer formed by α 1,4-linked chains of galacturonic acid that can be methylesterified and acetylated in specific carbons (C2-C3 and C6, respectively) [25]. Un-methylesterified homogalacturonan chains can form Ca^2+^ linkages, adopting an “egg-box” configuration, which forms a gel-like structure that strengthens the wall [26]. Other relevant pectic polysaccharides are rhamnogalacturonan I and II, which are branched oligosaccharides with a broad variety of monosaccharide compositions [27].

Hemicelluloses are a complex group of polysaccharides composed of a backbone of between 500–3000 sugar units (mainly d-xylose, d-glucose, and d-mannose) linked by β-(1→4) bonds. In addition to the linear backbone, hemicellulosic polysaccharides can be branched, and they are also subject to other modifications such as acetylation and methylation. Hemicelluloses associate with cellulose and pectins non-covalently, forming a network that contributes to increase the load-bearing capacity of cell walls. In addition, they also regulate cell expansion and prevent self-association of cellulose microfibrils [28]. Given the heterogeneous nature of this group, hemicellulose composition varies very much between different plant species. In *A. thaliana*, xyloglucan is the most abundant hemicellulosic polysaccharide in primary cell walls, whereas xylan is the major hemicellulose in secondary cell walls [29]. Both are synthesized in the Golgi bodies by glycosyltransferases and enzymes that add individual sugar decorations to the glucan backbone [28].

Callose and lignin are cell wall components that have been implicated both in defense responses and growth processes. Callose is a linear β-1,3-linked glucan polymer consisting of hundreds of d-glucose subunits, which is synthesized by plasma-membrane-localized callose synthases [30]. Callose is an early component of the primary cell wall separating the newly formed daughter cells after cell division and is frequently deposited at sites of pathogen infection, supposedly to reinforce the cell wall at an infection site and slow down the spread of the pathogen [31,32]. However, loss of callose synthase activity leads to increased pathogen resistance, apparently because it leads to enhanced salicylic acid (SA) levels and corresponding activation of defense responses [33]. These observations highlight how the cell wall can contribute to phytohormone-mediated defense responses. Lignin consists mainly of p-hydroxphenyl, guaiacyl, and syringyl units, which are produced by the phenylpropanoid and monolignol pathways [34]. The units are polymerized into higher-order structures embedded in the plant cell walls [35]. Lignin plays important roles during plant growth, exemplified by waterproofing the walls of vascular tissue cells or in the formation of the Casparian strip [36]. In addition to that structural role and similarly to callose, lignin is also deposited during plant–pathogen interactions to limit the spread of pathogens. Moreover, cell wall damage (CWD) caused by cellulose biosynthesis inhibition can also induce lignin deposition in primary cell walls [37,38]. Furthermore, targeted manipulation of lignin biosynthesis seems to result in the production of pectin-derived immune elicitors, inducing defense responses and growth inhibition [39]. These observations indicate that manipulation of cell wall components can result in adaptive changes or modifications of other components, and that these events are not restricted to particular cell wall types, not easily explained by redundancy, and could thus contribute to cell wall plasticity.

Most cell wall proteins are glycoproteins, which are divided into four groups: (i) glycine-rich proteins, (ii) proline-rich proteins, (iii) arabinogalactan-rich proteins, and (iv) hydroxyproline-rich glycoproteins or extensins [40]. They form, together with the polysaccharide components and lignin, complex three-dimensional networks. These give plant cell walls their unique characteristics in terms of mechanical properties, allowing controlled expansion while containing the high levels of turgor pressure prevalent in plant cells. Additionally, GPI-anchored proteins connected to the plasma membrane can also be considered cell wall proteins, since they are often connected to the cell walls as well, thus contributing to the formation of a plasma membrane–cell wall continuum [41]. Alteration in the functional integrity of cell walls or the disruption of the plasma membrane–cell wall continuum activates the CWI maintenance mechanisms. While the specific signals remain to be determined, the available evidence summarized below implicates mechano- and turgor-sensitive processes in combination with ligand–receptor interactions. 

## 3. Cell Wall Mechanoperception—Lessons from a Simple Organism

Plant and yeast cells, in contrast to animal cells, are both surrounded by sturdy extracellular matrices (cell walls) to protect the protoplast and contain the prevalent elevated turgor pressure levels. This influences how mechanical stimuli are perceived and necessitates the existence of dedicated CWI maintenance mechanisms in these organisms. These mechanisms were originally characterized in the yeast *S. cerevisiae*, which naturally resulted in a much more thorough understanding of their modes of action than what we know about the ones in plants, and were reviewed comprehensively some time ago [42]. Briefly, three mechanisms contribute to CWI maintenance in yeast by either directly detecting changes in the functional integrity of the cell wall or consequences thereof on the cell (distortion/displacement of the plasma membrane or changes in turgor pressure levels). Interestingly, for certain molecular components active in *S. cerevisiae*, homologs have been identified in plants, and the basic challenges regarding detection of mechanical stimuli and impairment of CWI are comparable in both organisms. Therefore, briefly surveying the knowledge available regarding yeast can facilitate dissection and help to understand the mechanisms active in plants.

The first mechanism for monitoring CWI in yeast involves a Ca^2+^ channel formed by Cch1 and Mid1, which is able to detect plasma membrane stretch caused by the combination of a weakened cell wall and the prevalent high turgor levels. Stretching leads to calcium influx, which regulates downstream responses via a signal transduction cascade involving calmodulin (Figure 1A) [42]. Expression in yeast of *A. thaliana*’s *MID1 COMPLEMENTING ACTIVITY 1* (*MCA1*) and *MCA2* partially rescues the phenotype of yeast strains deficient in Mid1, and their location in the plasma membrane is compatible with a role in mechanoperception [43,44]. In *A. thaliana*, MCA1 and MCA2 are localized at the plasma membrane, mediate Ca^2+^ influx triggered by mechanical stimuli and hypoosmotic pressure, and are necessary for responses to CWD (Figure 1B) [45,46]. Other mechanosensitive ion channels include those belonging to the MscS-like (MSL) family, which are similar to bacterial mechanosensitive channels of small conductance (MscS) and represent key players in controlling turgor levels in bacterial cells [47]. The *A. thaliana* genome contains 10 genes coding for MSLs, with MSL1 localized in mitochondria, MSL2 and 3 in plastids, and MSL4–10 at the plasma membrane. MSL2 and MSL3 (Figure 1B) can complement the lethality of an *Escherichia coli* mutant lacking MscS under hypoosmotic conditions, and are thus suggested to act as mechanosensitive channels [48], but their implication in mechanical-stimulus-induced release of Ca^2+^ has not been confirmed yet [45]. 

The second mechanism involves the cell-surface sensor kinase Sln1 [42]. This branch of the yeast turgor-pressure-monitoring mechanism activates the mitogen-activated protein kinase (MAPK) Hog1 and enables the yeast cell to detect hyperosmotic stress, i.e., shrinking of the plasma membrane or displacement of the membrane relative to the cell wall (Figure 1A). Expression of the *A. thaliana* genes *ARABIDOPSIS HISTIDINE KINASES* (*AHK*) *AHK1* and *AHK4* (also known as *CYTOKININ RECEPTOR1*, *CRE1*) in yeast strains impaired in Sln1 led to partial rescue of the mutant phenotype [49,50]. The location of AHK1 in the *A. thaliana* plasma membrane, like Sln1 in yeast, is compatible with its role as an osmoreceptor [51] (Figure 1B). The specific processes in *A. thaliana*, where mechano- and turgor sensors are involved in CWI maintenance, are addressed in detail in Section 4.

The third mechanism is capable of detecting modifications leading to changes in the mechanical characteristics of yeast cell walls. It involves five dedicated CWI sensors (Wsc1, 2, 3, Mtl1, and Mid2, Figure 1A), which are localized in particular plasma membrane locations [42,52]. They have highly mannosylated extracellular domains which seem to act as nanosprings, detecting mechanical deformation in the wall [53,54]. Conformational changes of the sensors upon wall deformation allow the interaction of their cytoplasmic domains with a small G protein, Rho1—considered to be the master regulator of CWI signaling—and guanosine nucleotide exchange factors (GEFs). This interaction activates downstream responses via protein kinase c (Pkc1) and MAPKs (Figure 1A) [42]. These responses include changes in cell wall biosynthetic activity and organization of the cytoskeleton [42]. While no homologs for the yeast CWI sensors have been identified in plants, *Catharanthus roseus* receptor-like kinases (*Cr*RLKs) like THESEUS1 (THE1) or FERONIA (FER) have been implicated in plant CWI maintenance (Figure 1B) as sensors, as reviewed recently [55]. Receptor kinases belonging to the *Cr*RLK family are found throughout the plant kingdom and have been implicated in gametophytic development, root hair growth, abiotic stress response, and plant–pathogen interactions, all processes where CWI needs to be monitored and actively maintained [56,57,58]. Both THE1 and FER are plasma-membrane-localized and have extracellular malectin domains, which possibly interact either directly or via other proteins with cell wall components, while their kinase domains reside on the cytoplasmic side. Moreover, downstream signaling events also include GEFs and plant RHO-related GTPases [59], similar to the system in yeast. Details on *Cr*RLK’s role in plant CWI monitoring systems, together with other receptor-mediated CWI monitoring systems, are provided in Section 5. 

These branches converge in the control of the transcription factor Skn7, which coordinates the expression of CWI- and osmotic-pressure-related genes (Figure 1A). The combination of signals deriving from the three mechanisms provides yeast cells with a high-resolution, three-dimensional map of the state of the cell wall–plasma membrane continuum and allows tightly controlled, specific changes in protoplast and/or cell wall metabolism, enabling adaptation to environmental changes or developmental processes (Figure 1A). Adaptation to the latter is exemplified by the CWI checkpoint, which directly influences yeast cell cycle progression [60]. A similar situation seems to exist in *A. thaliana*, where cellulose biosynthesis inhibition leads to an arrest of cell cycle activity, which is sensitive to osmotic manipulation [61]. Currently, there is no knowledge regarding the transcriptional machinery controlling the responses to CWI impairment in plants.

## 4. Perception of Turgor Pressure and Mechanical Stimuli in the Plant Cell Wall–Plasma Membrane Continuum

Turgor pressure perception and maintenance via mechanosensing, have been identified as key processes underlying plant development and interaction with the environment. Despite this importance, our understanding of the responsible mechanisms is still surprisingly limited [62,63,64]. Turgor pressure levels are monitored by two main families of proteins localized in different subcellular compartments, including chloroplasts and the plasma membrane: histidine kinases and channel complexes. Furthermore, new evidence from observations in *A. thaliana* seedlings points to a third mechanosensitive system, in which microtubules acting as tension sensors regulating developmental processes contribute also to mechanosensing in plant cells [65,66].

A well-studied case, representative of the histidine kinase group of turgor pressure sensors, is AHK1, which has been implicated in perception of both hyperosmotic stress and cytokinins (Figure 1B) [51,67]. However, the available evidence suggests that the precise roles of the AHKs remain to be determined, since AHK1 is not required for induction of the phytohormone abscisic acid (ABA) in response to drought, a key regulator of hyperosmotic stress responses [68]. In the case of channel complexes, examples include the *A. thaliana* plasma-membrane-localized Ca^2+^ channels REDUCED HYPEROSMOLALITY INDUCED CA^2+^ INCREASE 1 (OSCA1.1) and 1.2, which are mechanically activated and required for perception of hyperosmotic stress (Figure 1B) [69,70,71,72]. Another example of proteins involved in response to osmotic stress are mechanosensitive ion channels MSL2, 3, and 10 (Figure 1B). MSL2 and 3 mediate adaptation to hypoosmotic stress and contribute to CWD-induced production of jasmonic acid [48,73]. In addition, MSL10 is localized in the plasma membrane and is required for induction of hypo-osmosis-induced responses such as Ca^2+^ influx, production of reactive oxygen species (ROS), and expression of mechanosensitive genes [74]. MSL10 activity seems to be regulated by its phosphorylation state and modulates cell-swelling-induced cell death (Figure 1B). Similarly, the Ca^2+^-channel MCA1 has been implicated in mechanosensing, hypoosmotic stress perception, and induction of jasmonic acid (JA), SA, and lignin production in response to CWI impairment [43,44,73,75,76] (Figure 1B). The involvement of MCA1 in different processes illustrates how closely interconnected mechanosensing, turgor monitoring, and CWI maintenance are in plants (Figure 1B). More importantly, this interconnection may lead to systemic redundancy beyond that generated by gene families and could explain the limited mutant phenotypes frequently observed with single gene knockouts [4]. In this context, results from experiments with *A. thaliana* seedlings exposed to cellulose biosynthesis inhibitors (like isoxaben, affecting only expanding cell walls), enzyme preparations containing cell-wall-degrading activities (driselase), and osmotica are relevant [73,77]. The results indicate that in *A. thaliana* seedlings, induction of JA, SA, lignin production, and defense gene expression are osmo-/mechanosensitive. Intriguingly, THE1 was required for JA production induced by both cell-wall-degrading enzymes and isoxaben, suggesting that THE1 activity is required in cellular processes affecting CWI in general. The effects observed with these treatments are complemented by results from genetic studies using 25 different genotypes, which found that genes involved in cell wall signaling and hypoosmotic stress perception (incl. MCA1, MSL2, MSL3, and MSL10) contribute to responses induced by CWD caused by cellulose biosynthesis inhibition [73]. Interestingly, these results provided further evidence for a role in cell wall integrity signaling for other molecular components such as the receptor-like kinase FEI2 (Figure 1B). In contrast, genes required for responses to hyperosmotic stress were not required. These results support the concept that CWD-induced weakening of cell walls may have effects comparable to, but more pronounced than, those caused by hypoosmotic stress, i.e., cell expansion or swelling. In secondary cell walls, a different mechanism could be active in response to CWD, since manipulation of lignin results in the release of pectin-derived small molecules, activating defense responses and influencing resistance to pathogen infection [39]. These observations serve as a reminder that secondary cell walls need to meet different biological requirements and have different mechanical characteristics than primary walls, and therefore CWI impairment could also be detected in a different manner. 

## 5. Receptor-Mediated Cell Wall Integrity Monitoring

Recent years have seen a dramatically increased interest in the mechanisms underlying cell wall signaling and CWI maintenance, for several reasons. CWI maintenance seems to be an important element of cell wall plasticity, capable of neutralizing targeted manipulation of cell wall metabolism [3,4]. The contributions of CWI maintenance signaling to biotic and abiotic stress responses have become more obvious [57,58,73,78]. Elements of the CWI maintenance mechanism have been implicated in regulation of cell elongation and gametophytic development [79,80]. Simultaneously, evidence has accumulated that CWI signaling is influencing the performance of food crops, highlighting the commercial relevance of improved knowledge regarding the mode of action of this mechanism [4,81]. Members of several protein families have been implicated in CWI maintenance signaling processes [82]. Here, we focus only on certain relevant aspects of two *Cr*RLKs (FER and THE1), because they form key elements in the CWI maintenance mechanism, knowledge regarding their functions has increased pronouncedly in recent months, and the protein family itself has been recently and very competently reviewed [55]. THE1 was originally identified through its function in regulating responses to CWI impairment caused by reduction of cellulose production in elongating cell walls [83]. However, THE1 is also required for resistance against *B. cinerea* infection and for production of JA/SA/lignin induced by cell-wall-degrading enzymes, suggesting that THE1 function is more general [58,73]. Genetic analysis found that THE1 acts upstream from MCA1 in regulating JA/SA/lignin production induced by cellulose biosynthesis inhibition (Figure 1B) [73]. Rapid-alkalinization factor (RALF) peptides have been proposed to be ligands for *Cr*RLKs, and indeed, THE1 interacts with RALF34, hinting at possible ways of perceiving signals [79]. THE1 also interacts with GUANINE EXCHANGE FACTOR4 (GEF4), indicating a possible signaling pathway to downstream targets [58] (Figure 1B). While the downstream signaling processes remain to be elucidated, the pH-dependent nature of the RALF34/THE1 interaction supports the notion of additional pH-based regulatory mechanisms controlling the activity of THE1-based signaling (Figure 1B).

FER is involved in a multitude of processes beyond CWI monitoring, including innate immunity, stress response, cellular growth, morphogenesis, and fertilization [57,84,85,86,87,88]. FER forms part of the signaling heterocomplexes in the plasma membrane that mediate signal perception, suggesting that its description as a scaffold that integrates different signals to initiate specific downstream responses might be most appropriate. For example, FER is involved, together with LRE-like GPI-AP1 (LLG1) and BRASSINOSTEROID INSENSITIVE 1–ASSOCIATED KINASE 1 (BAK1), in the perception of RALF23 (Figure 1B), and this event is required for an efficient activation of plant immunity [89,90]. In addition to RALF peptides, malectin domains have also been suggested to bind carbohydrates [91]. Therefore, it has been proposed that FER could also perceive ligands derived from pectic polysaccharides. Moreover, FER ligands, either RALF peptides or pectin fragments, could be released or activated in response to CWI impairment. FER interaction with leucine-rich-repeat extensins (LRXs) has been proposed as an essential step in the detection of CWI impairment [57] (Figure 1B). A recent work showed that LRX proteins can interact both with FER and the cell wall, coordinating responses such as cell wall loosening and decrease of pH with an increase in vacuolar size, which regulates cell expansion [92]. It remains to be clarified whether FER binds to pectins only in vitro [93] or also in vivo. However, it has been demonstrated that FER regulates the esterification state of pectins during gametophytic development to prevent double fertilization [80]. This prevention also requires nitric oxide (NO) release, possibly generated by NITRATE REDUCTASE1 NITRATE REDUCTASE2 (NIA1 NIA2) [80]. Intriguingly, CWI impairment caused by cellulose biosynthesis inhibition induces the production of ROS, while responses like JA/SA/lignin production are NIA1 NIA2-dependent [61,94] (Figure 1B). FER acts through a guanine nucleotide exchange factor–plant RHO-related GTPase (GEF-ROP/RAC) pathway to regulate ABA-based signaling while also being regulated in turn by ABA, hinting at the existence of regulatory feedback loops that may also affect turgor pressure levels in plant cells [95,96] (Figure 1B). Since FER activity is modified by interaction with RALF1, the available data suggest that FER could form a node in a regulatory network coordinating CWI with ABA-based signaling. This notion is further supported by the observation that RALF1–FER interaction leads to increased FER phosphorylation, which in turn inhibits the activity of ARABIDOPSIS H^+^ ATPase2 (AHA2) [86,97] (Figure 1B). The FER–ABA regulatory feedback loop is not the only one described, which explains to some extent the multitude of functions ascertained to FER. For example, FER also interacts with EBP1 to control cell growth while in parallel influencing flowering time [98,99].

## 6. Perception of Cell Wall Damage: Wall-Derived Damage-Associated Molecular Patterns

One approach used by organisms to detect tissue damage is the perception of DAMPs. DAMPs are generally defined as endogenous molecules that activate and modulate the innate immune responses after being passively or actively released into the extracellular space. The perception of DAMPs by pattern-recognition receptors is a cornerstone of pattern-triggered immunity in both animal and plant innate immunity [100,101]. Plant cell walls are a potential major source of DAMPs, which can be derived from any cell wall component [102]. These alterations of CWI trigger specific defensive responses, including elevations in cytoplasmic [Ca^2+^], ROS production, activation of MAPK signaling pathways, SA and JA production, callose and lignin deposition, and changes in gene expression, considered hallmarks of classical pattern-triggered immunity [102]. 

To date, only a few cellulose-derived DAMPs have been identified. The disaccharide cellobiose is one such cellulose-derived molecule that exhibits DAMP activity. Cellobiose treatment established immunity signaling responses in *A. thaliana* seedlings [103]. In more recent work, other cello-oligomers with a degree of polymerization between three and seven were also shown to trigger increases in cytoplasmic [Ca^2+^]. Amongst these, cellotriose was the most active compound, also triggering ROS production in *A. thaliana* roots, in contrast to cellobiose [104]. However, the pathogen-recognition receptor(s) that detect the presence of these cello-oligosaccharides have not been identified yet. 

Pectins represent a major source of cell-wall-derived DAMPs. A classic example is oligogalacturonides (OGs), linear molecules of about 10 to 16 α-1,4-d-galacturonosyl residues derived from homogalacturonan in primary cell walls. Recent studies have shown that they are released in vivo during interaction of *A. thaliana* with the fungus *Botrytis cinerea* [23]. OGs activate the same immunity responses as other elicitors [105,106,107,108] and seem to be perceived by WALL ASSOCIATED KINASE1 (WAK1) [109] (Figure 1B). This kinase belongs to a large family of pattern-recognition receptors with 22 members that have all epidermal-growth-factor-like ectodomains [110] (Figure 1B). The MAPK kinase kinases ARABIDOPSIS NPK1-RELATED PROTEIN KINASE (ANP) 1, 2, and 3 are also required for the transduction of OG-induced signals and downstream responses [61,111,112] (Figure 1B). In parallel, OGs also trigger inhibition of hormone-induced gene expression [112,113], highlighting the role of OGs in balancing stress responses and development. Shorter OGs can induce most immunity responses, but not ROS production, suggesting that OGs with different degrees of polymerization may trigger different responses [114]. However, given the complexity and availability of pectic polysaccharides in general, other pectin-derived DAMPs could also contribute to CWD perception. The recent discovery that pectin-enriched cell wall fractions activate most immunity responses supports this hypothesis [115]. Intriguingly, cell wall fractions from plants mutated for ARABIDOPSIS RESPONSE REGULATOR 6 (ARR6), implicated in cytokine perception, trigger enhanced immunity responses [115]. These results, together with the observation that pectin-derived signaling molecules modulate responses to lignin modification [39], suggest that pectins could be an important source of cell wall DAMPs in both primary and secondary cell walls. Therefore, a targeted, in-depth analysis of their respective activities will probably lead to exciting new insights into the signaling processes regulating responses to CWI impairment. 

Hemicellulose-derived DAMPs have remained elusive, but some pioneering work in pea and bean has suggested a role for xyloglucan-derived hepta- to nona-saccharides in regulating auxin-induced growth [116,117]. While recent work suggested that xyloglucan oligomers obtained by enzymatic extraction and purification from apple can trigger pattern-triggered immunity responses in *A. thaliana* and grapevine [118], the molecular components responsible remain to be characterized. 

Callose is also a component of fungal cell walls [119]. Therefore, callose-derived oligosaccharides can be considered to be both DAMPs and fungal elicitors. Recently it was shown that β-1,3-linked glucans of different degree of polymerization are indeed perceived by both monocots and dicots, activating different immunity responses [119,120]. In *A. thaliana*, the six-monomer β-1,3-glucan laminarihexaose is perceived by pattern-recognition receptors belonging to the LysM family: CHITIN ELICITOR RECEPTOR KINASE 1 (CERK1), LYSIN MOTIF-CONTAINING RECEPTOR-LIKE KINASE 4 (LYK4), and LYK5 [119] (Figure 1B). In contrast, in the case of *Nicotiana benthamiana*, the equivalent LysM pattern-recognition receptors were not required to detect β-1,3-glucan oligosaccharides [120]. 

Cell-wall-derived DAMPs activate classic pattern-triggered immunity responses and coordinate important physiological processes such as defense, growth, and development in plants, all of which require a functional cell wall. This is exemplified by cell wall modifications that lead to release of cell-wall-derived DAMPs and causing changes in pathogen resistance [102,118]. Likewise, DAMP perception has also been linked to the perception of abiotic stresses, exemplified in *A. thaliana* and rice plants by the increased expression of *WAK*s in response to metals in soil [121,122]. However, the mechanisms responsible for the coordination of cell wall composition and structure, release of DAMPs, activation of defense responses, changes in cell wall and cellular metabolism, growth rates, and resistance to different biotic and abiotic stresses are still not well understood. 

## 7. How Do the Different Sensing Mechanisms Contribute to Cell Wall Integrity Monitoring?

The selective overview presented above highlights the recent progress made in our knowledge regarding plant CWI maintenance mechanisms and the increased attention the topic is attracting in the community, as evidenced by the large number of recent publications. While our understanding of the detailed mode of action of the CWI maintenance mechanism is still limited, the information available is sufficient to propose a concept that can be used to guide future experiments investigating the mode of action of the mechanism responsible for CWI maintenance in primary cell walls (Figure 2). 

The space encompassing the apoplast and plasma membrane can be considered as a continuum where any stimulus indicative of CWI impairment is perceived and translated to downstream responses involving changes in cell wall and cellular metabolism [7,123]. The specific nature of the initial stimulus remains to be determined, since the available information supports different possible scenarios. It could consist of cell-wall-derived fragments [102] or involve the detection of distortion of the plasma membrane and/or displacement of the membrane versus the cell wall [73] and/or monitoring of turgor levels, as well as a combination of fragment perception and plasma membrane modification (Figure 2). 

Bearing in mind these considerations, the candidate proteins currently implicated in CWI maintenance would enable detection of all three types of stimuli. Examples of receptors capable of binding cell wall fragments and activating downstream responses are WAK1, LYK4-5, and CERK1, possibly transducing signals through ANPs and MAPKs to downstream response mediators [61,109,119] (Figure 2A). Plasma-membrane-localized channel proteins like AHK1, OSCA1.1, OSCA1.2, MCA1, MCA2, and MSL10 enable detection of plasma membrane stretch, while chloroplast-localized MSL2 and MSL3 allow monitoring of turgor levels independently of the state of the plasma membrane, since their reduction could also be indicative of a weakened cell wall [43,48,51,74,75] (Figure 2B,C). Studies using calcium signaling inhibitors further support the notion that calcium-based signaling processes are involved [76]. The characterization of plasma-membrane-localized FER suggests that it can bind pectin-derived cell wall fragments or bind to cell walls directly or indirectly via LRX proteins [57,92]. This would allow detection of CWD and/or displacement of wall versus membrane (Figure 2B,C). With the binding specificity of FER and THE1 for RALF peptides apparently changing in a pH-dependent manner, RALFs interacting with more than one *Cr*RLK, and FER regulating AHA2 activity, additional regulatory levels exist, which suggests that fine-tuning of the signaling activity and coordination between different *Cr*RLKs may occur [79,86,97]. Signals from THE1 and FER are possibly relayed via NO- and/or ROS-based signaling cascades [73,80,94]. THE1 is required for JA/SA production in response to CWI impairment (Figure 2C), while loss of FER actually leads to enhanced JA/SA production [73]. This observation suggests that *Cr*RLKs can have opposite functions in the same process, and raises the obvious question of how the differences in activity are generated. While FER regulates ABA-based signaling processes via a GEF-ROP/RAC pathway and ABA in turn can regulate FER activity (Figure 2B,C), it remains to be determined whether THE1 has also a function in ABA regulation, and to what extent GEF-based signaling processes relay THE1-derived signals [95,96]. Moreover, ABA-based processes regulate a large number of general stress responses and they are also key regulators of turgor pressure [46,124]. Intriguingly, cellulose levels are reduced in *aba deficient 1* (*aba1*) mutant plants, suggesting that there may be a role for ABA signaling in regulation of cellulose production, or that changes in turgor pressure may affect cellulose production and cell wall sturdiness [125,126]. Most of the signaling components mentioned here are members of gene families, which hold the obvious potential for redundancy and thus possibly explain the often subtle knockout phenotypes observed. However, the phenotypes observed cannot always be explained by redundancy within individual gene families, hinting at redundant organization of the CWI maintenance mechanism and the plasticity in place [4]. 

## 8. Perspective: Open Questions and Future Challenges 

In this short review, the focus was the CWI maintenance mechanism active in primary cell walls in *A. thaliana* and the contributions of mechanosensing to it. The summary condensed the significant progress made in recent years and provides cues to improve our current conceptual framework, thus allowing us to dissect the processes responsible in a hypothesis-driven manner. It highlighted that mechano-/turgor sensing are intricately involved in CWI maintenance. The next steps will be to determine to what extent CWI maintenance exists in both crop and non-model plants and investigate CWI maintenance processes in secondary cell walls [4]. Our knowledge is very limited, but it suggests that interesting research opportunities exist and will feed scientific discussion in the future [127,128]. While research into food and energy crop plants has the obvious attractions of potentially leading to results with translational potential, investigating plants surviving in extreme environments may yield results with more pronounced impact in the long term [129]. 

## Figures and Tables

**Figure 1 plants-09-00574-f001:**
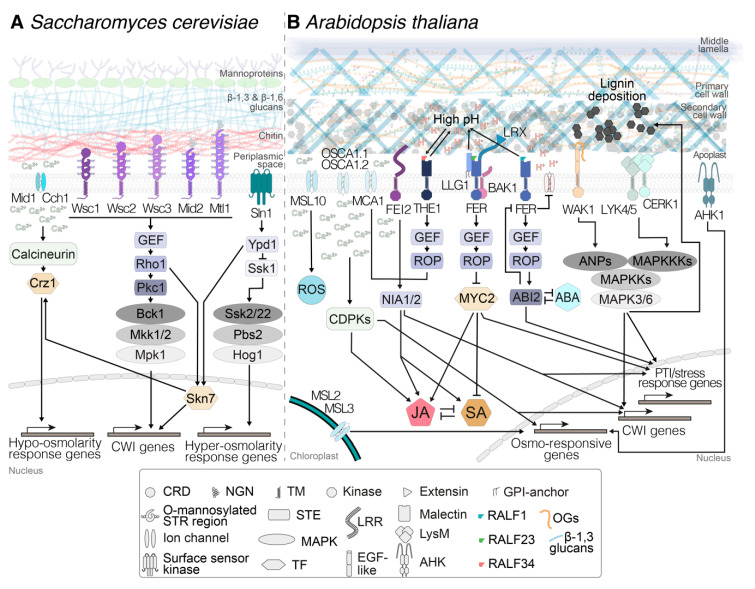
Comparison between cell wall integrity (CWI) maintenance mechanisms in (**A**) *Saccharomyces cerevisiae* and (**B**) *Arabidopsis thaliana.* In both organisms, mechanosensitive ion channels and receptors trigger signal transduction processes involving Ca^2+^ influx into the cytoplasm and the activation of cascades including calcium-dependent protein kinases (CDPKs), guanosine nucleotide exchange factors (GEFs) and mitogen-activated protein kinases (MAPKs) that eventually activate transcription factors. In *A. thaliana*, the processes are regulated in a more intricate manner, exemplified by the complex interconnected networks regulated by jasmonic acid (JA), salicylic acid (SA), and abscisic acid (ABA). The processes enable the regulation of gene expression in a tightly controlled and highly adaptive manner, allowing specific changes in cell wall and cellular metabolism to maintain cell wall integrity. Arrows are connecting elements belonging to the same pathway. ROS: reactive oxygen species; PTI: PAMP-triggered immunity; CRD: cysteine-rich domain; NGN: N-glycosilated asparagine; TM: transmembrane domain; GPI: glycosylphosphatidylinositol; STR: serine-/threonine-rich; STE: signal transduction element; LRR: leucine-rich repeat domain; LysM: lysin-motif-containing ectodomain; RALF: rapid alkalinization factor peptide; OGs: oligogalacturonides; TF: transcription factor; EGF-like: epidermal-growth-factor-like ectodomain.

**Figure 2 plants-09-00574-f002:**
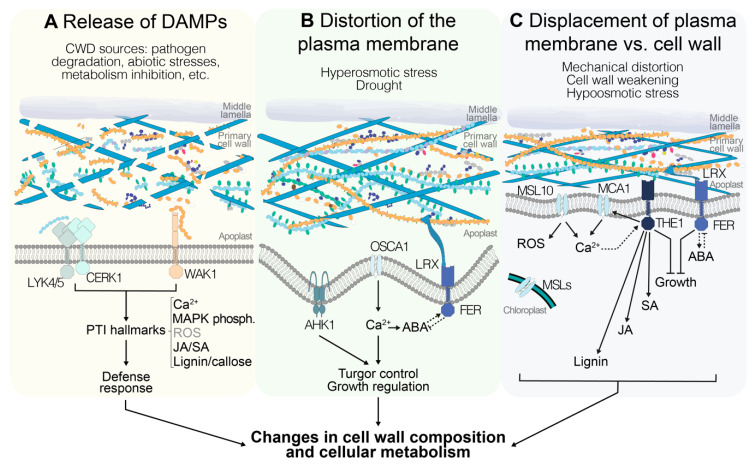
Different stimuli could indicate alterations in cell wall integrity (CWI) in *Arabidopsis thaliana*, and are perceived through different signaling pathways. (**A**) Release of cell wall fragments, also known as damage-associated molecular patterns (DAMPs), is interpreted by the plant as the result of cell wall damage (CWD) that can derive from biotic and abiotic stresses as well as endogenous processes. DAMP perception by pattern-recognition receptors activates typical immunity responses including increases in cytoplasmic [Ca^2+^], phosphorylation of mitogen-activated protein kinases (MAPKs), production of jasmonic acid(JA)/salicylic acid (SA), reactive oxygen species (ROS), and lignin, and callose deposition. (**B**) Distortion of the cell wall–plasma membrane continuum occurs in response to plasma membrane shrinkage during hyperosmotic stress (drought). Mechanosensitive ion channels mediate Ca^2+^ influx into the cytoplasm, leading to the activation of signal transduction pathways. Moreover, interactions between abscisic acid (ABA) and FER modulate growth in response to the state of turgor pressure. (**C**) If the plasma membrane is stretched, either by a weakened cell wall or as result of hypoosmotic stress, mechanosensitive ion channels are activated and [Ca^2+^] in the cytoplasm is increased. Several of these channels are also required for the production of ROS. Moreover, THE1-mediated signaling modulates CWD-induced lignin and JA/SA production, and together with FER leads to growth arrest until CWI is recovered. The pathways in **A**, **B**, and **C** eventually lead to a series of changes in cell wall composition and cellular metabolism, enabling the plant to maintain CWI in response to different challenges.

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
