# Peer review of "The Role of Mechanoperception in Plant Cell Wall Integrity Maintenance"

_plants, 2020, doi:10.3390/plants9050574_

Round 1

Reviewer 1 Report

The review article "The role of mechano-perception in plant cell wall 2 integrity maintenance" is an interesting and comprehensive update on this subject, long time after the previous one by T. Hamann (doi:10.1093/pcp/pcu164). It is well-written, in general, and should be published. However there are some points to be amended (see also attached PDF with annotations).

  1. There are too many abbreviations that make reading of the text really confusing. I expect the authors to keep them to a minimum and full-spell all the rest.
  2. There is one error concerning the work of Haas et al. concerning pectin nanofilaments: Nanofilaments have been shown in anticlinal but not periclinal walls.
  3. A slight English polishing is also required.
  4. It would be very helpful to include a diagramm illustrating the directions of further research, based on all the information reviewed in this manuscript.

Author Response

Responses to reviewer 1:

There are too many abbreviations that make reading of the text really confusing. I expect the authors to keep them to a minimum and full-spell all the rest.

We agree with the referee that this can be challenging. We have therefore reviewed the usage of abbreviations and reduced them as much as possible.

There is one error concerning the work of Haas et al. concerning pectin nanofilaments: Nanofilaments have been shown in anticlinal but not periclinal walls.

We have corrected this mistake.

A slight English polishing is also required.

We have performed further proofreading and examined carefully opportunities to improve the quality of the English.

It would be very helpful to include a diagram illustrating the directions of further research, based on all the information reviewed in this manuscript.

We have considered this suggestion carefully but bearing in mind that that the perspective section is very short and not easily translated into such a diagram we think such a diagram will not be helpful for the reader.

Minor comments:

Line 25 Corrected

Line 51 Corrected

Line 58: cell walls contain the high turgor pressure levels.

Line 60: we do not wish to focus solely on cellulose here but also like to include other polysaccharides found in 2ndary cell walls, therefore we are not using only “cellulose”.

Line 70: Has been modified.

Line 80: we have used in the revised version (where appropriate) Arabidopsis.

Line 89: has been corrected with “anticlinal” wall.

Line 93: has been rewritten in response to comments by reviewer 2.

Line 129: has been corrected to “waterproofing”.

Lines 141 – 144: Wording has been modified where appropriate and sentence rephrased.

Line 154: Corrected

Lines 157-158: Has been rephrased.

Lines 171-172: Corrected

Line 256: has been rephrased and the abbreviation “DAMP” has been defined before (line 45)

Reviewer 2 Report

Major Comments

1. Multiple sentences in the first paragraph of section 2.1 are directly or partly copied from reference 7 and should be re-worded to avoid the appearance of plagiarism. This section in general seems a bit long and detailed and could be edited to include only the information needed to understand what comes later.

2. The yeast/Arabidopsis comparison has been made before by these authors (for example Hamann (2015) in Phytochemistry; Hamann  2015 in Plant Cell Physiol). This paragraph is very long; perhaps the section from line 175-187 could be a separate paragraph that systematically compares what is known in Arabidopsis to the three systems from yeast (cell wall/membrane tension/hyper-osmolarity), and then draws out the any implications to make it clear to the reader what is useful about the comparison.

3. Line 220-21 I am not sure that MSL2/3 and MSL10 are implicated in hyper-osmotic stress responses, only hypo-osmotic.

4. Line 395: The proposed role of MSL1 and MSL2 in turgor perception from the chloroplast is intriguing but the exact idea being proposed is unclear from the statement “since their reduction could be indicative of a weakened cell wall”. Note that MSL1 is localized to mitochondria; perhaps MSL3 is meant. Also, chloroplast-associated channels are shown in the figure to modulate calcium signaling; this idea should be cited or explained.

5. Figure 2C does not reflect the statement made here (line 334) or elsewhere (Englesdorf 2018) that MCA1 functions downstream of THE1 in CWD perception.

Minor comments

  1. In Figure 1. Snl1 needs to be corrected to Sln1. The figure resolution, especially in apoplast, is low
  2. Some references are missing:

Ref for Sln1 (line 166).

Ref for line 178 “CRE1 can compensate to some extent for loss of Sln1”.

Ref for “The responses… organization of the cytoskeleton.” (line 161)

Ref for “Pectin … 50% of the primary cell wall” (line 85-6)

Ref for “This … possibly generated by … NIA/2 (line 359-361)

  1. The statement in Line 231-2 is intriguing but needs additional explanation

  2. Line 227,228, “ Its´ “ should be “Its”.

Author Response

Responses to reviewer 2:

Major Comments

  1. Multiple sentences in the first paragraph of section 2.1 are directly or partly copied from reference 7 and should be re-worded to avoid the appearance of plagiarism. This section in general seems a bit long and detailed and could be edited to include only the information needed to understand what comes later.

We were rather surprised by this observation and went back to the text in question. Several of the sentences seem simply pretty generic and frequently used to us, which therefore creates the wrong impression of plagiarism. We have now rephrased the section in question and shortened the part on pectins to address the concerns of the referee regarding section length in general.

  1. The yeast/Arabidopsis comparison has been made before by these authors (for example Hamann (2015) in Phytochemistry; Hamann  2015 in Plant Cell Physiol). This paragraph is very long; perhaps the section from line 175-187 could be a separate paragraph that systematically compares what is known in Arabidopsis to the three systems from yeast (cell wall/membrane tension/hyper-osmolarity), and then draws out the any implications to make it clear to the reader what is useful about the comparison.

We have reorganised and shortened the paragraph where in our eyes possible and do hope this addresses the concerns of the referee.

  1. Line 220-21 I am not sure that MSL2/3 and MSL10 are implicated in hyper-osmotic stress responses, only hypo-osmotic.

This is a typo, MSL2, 3 and 10 have been implicated in hypo-osmotic stress. The mistake has been corrected.

  1. Line 395: The proposed role of MSL1 and MSL2 in turgor perception from the chloroplast is intriguing but the exact idea being proposed is unclear from the statement “since their reduction could be indicative of a weakened cell wall”. Note that MSL1 is localized to mitochondria; perhaps MSL3 is meant. Also, chloroplast-associated channels are shown in the figure to modulate calcium signaling; this idea should be cited or explained.

Another typo, obviously it should be MSL2 and 3 not MSL1. This has been corrected.

  1. Figure 2C does not reflect the statement made here (line 334) or elsewhere (Englesdorf 2018) that MCA1 functions downstream of THE1 in CWD perception.

We have amended Figure 2c to ensure it correctly illustrates the statement made here and in Engelsdorf et al., 2018.

Minor comments

  1. In Figure 1. Snl1 needs to be corrected to Sln1. The figure resolution, especially in apoplast, is low

The typo has been corrected and figure resolution / contrast has been adjusted.

  1. Some references are missing:

Ref for Sln1 (line 166).

A reference has been added (41).

Ref for line 178 “CRE1 can compensate to some extent for loss of Sln1”.

References have been added (48, 49).

Ref for “The responses… organization of the cytoskeleton.” (line 161)

A reference has been added (41).

Ref for “Pectin … 50% of the primary cell wall” (line 85-6)

A reference has been added (18).

Ref for “This … possibly generated by … NIA/2 (line 359-361)

A reference has been added (79).

  1. The statement in Line 231-2 is intriguing but needs additional explanation

This statement is meant as summary of the information presented in the preceding paragraphs regarding the involvement of MCA1 in different biological processes. Therefore, we do not see any need for additional explanations.

  1. Line 227,228, “ Its´ “ should be “Its”.

Corrected.
